# Maternity care providers' experiences of work-related serious events (MATES): An International survey

**Maria Healy** [1]*, **Patricia Leahy-Warren**[2], **Jean Calleja-Agius**[3], **Neville Calleja**[4], **Joan Lalor**[5], **Eleni Hadjigeorgiou**[6], **Marianne Nieuwenhuijze**[7]

1 School of Nursing and Midwifery, Queen's University Belfast, Lisburn, Northern Ireland, 2 School of Nursing and Midwifery, University College Cork, Cork, Ireland, 3 Department of Anatomy, Faculty of Medicine and Surgery, University of Malta, Msida, Malta, 4 Department of Public Health, Faculty of Medicine and Surgery, University of Malta, Msida, Malta, 5 School of Nursing and Midwifery, Trinity College Dublin, Dublin, Ireland, 6 Department of Nursing, School of Health Science, Cyprus University of Technology, Limassol, Cyprus, 7 Research Centre for Midwifery Science, Zuyd University & CAPHRI, Maastricht University, Maastricht, The Netherlands

* maria.healy@qub.ac.uk

## Abstract

Internationally, many women experience physiological childbirth with positive experiences and good health outcomes for them and their baby. For some, due to health complications and context of childbirth they may experience or be perceived as having had a traumatic birth. Ultimately, whether an individual experiences an event as traumatic or not, is a personal experience. Caring for women who experience their birth as traumatic can be challenging. The risk of exposure to a traumatic birth event(s) as part of maternity care providers (MCPs) work, ranges from 67% to 90%. Thereby to support MCPs to provide quality maternal and newborn care, it is important to explore the impact of work-related events. An anonymised online survey relating to MATernity serious EventS (MATES) was developed utilising validated instruments and hosted on Qualtrics XM. Following ethical approval, the questionnaire was disseminated internationally across 33 countries between 1st July and 31st December 2022 via social media and the COST Action DEVOTION (CA18211 www.ca18211.eu) network. In total, 579 MCPs responded with a wide age range and years of experience. Descriptive and inferential statistical analyses were performed, including univariate and multivariate linear regression. Data analyses and management were undertaken using SPSS v.20 and two-sided significance tests were applied (α 0.05). The findings suggest that MCPs are exposed to a large variation of serious events and continue to be intensely affected, up to the present day. Events ranged from stillbirth, neonatal death, maternal death, severe or life-threatening maternal or infant incidents and violence and aggression from women or family member. Institutional support for staff is limited, and when available, seldom used. Family and friends were relied on for support, but this does not appear to be associated with MCPs experiencing less secondary traumatic stress. Subsequently MCPs were absence from work through sickness (22%), changed their professional allocation (19%) and seriously considered leaving (42%). Moreover, many reported low to moderate compassion satisfaction and burnout (65–80%).

**Data availability statement:** All relevant data are within the paper and its Supporting Information files.

**Funding:** The author(s) received no specific funding for this work.

**Competing interests:** The authors have declared that no competing interests exist

**Abbreviations:** MCP, Maternity care provider; COST, European cooperation in science and technology.

With international scarcity of MCPs, the impact of these events seem to contribute to this shortage. Effective support for MCPs is required if staff are to be recruited and retained.

## Introduction

A serious event can be characterised along a continuum from an unpleasant occurrence to a very serious incident which is considered traumatic for the individual(s). Such events are not often due to a major disaster but usually an individual incident that occurs without warning, or incremental symptoms that could be resolved in a timely manner. The World Health Organization (WHO) stresses that a positive childbirth experience should be the ultimate goal of maternity care for all women, even when medical interventions are necessary [1]. However, serious complications can occur throughout the perinatal period, with most happening during labour and birth [2]. Due to a negative unfamiliar situation or an emergency during childbirth, individuals can experience increase in anxiety and pain levels in sequence with a decrease of oxytocin release due to stress defence response [3]. Each individual will interpret the situation differently, including the woman, as to whether their experience was traumatic or not. Some maternity care providers (MCP), such as midwives, obstetric nurses and doctors can experience loss of control and fear when managing or witnessing a serious event. This can make MCPs feel helpless and afraid for the well-being and indeed the life of the baby, the women and birthing people. The American Psychiatric Association Diagnostic and Statistical Manual's post-traumatic stress criteria can therefore be utilised in situations where serious events during labour occur [4].

Worldwide, the incidence of MCPs experiencing traumatic birth event(s) as part of their work ranges from 67% to 90% [5–8]. MCPs who experienced a traumatic event reported stress, intense feelings of fear, guilt, depression, withdrawal helplessness or panic in connection with a severe event [9,10]. Inherent within the MCPs remit is their close partnership relationship with women during the perinatal period. This close relationship can enhance intense emotions, which raise the risk and intensity of emotional stress in the event of an unfavourable or traumatic scenario [11].

MCPs have been found to develop secondary traumatic stress disorder as a result of this [12]. Secondary traumatic stress develops because of witnessing the trauma and manifests symptoms akin to those of post-traumatic stress disorder (PTSD), including depressing dreams, persistent anxiety, anger, depression, hopelessness, irritability, discomfort associated with remembering the event, difficulty focusing, and insomnia [13]. Other studies found that MCPs who experienced traumatic maternity events reported compassion fatigue, burnout, and their professional quality of life was negatively affected [14–18]. The latter study (Katsantoni *et al.* [18] identified the majority of respondents, which included certified nurses, midwives, and nurse/midwives in 3 public hospitals (80% of n = 121) were in the high-risk category for compassion fatigue and secondary traumatic stress (73.9%), while only 19.8% and 5% of nurses expressed high potential for compassion fatigue and burnout, respectively. They concluded that awareness of the factors associated with compassion fatigue may help MCPs to prevent or offset the development of this condition [18]. Frequently maternity care providers who witnessed traumatic events and experienced intense adverse emotions did not receive adequate support, either formally or informally [9,19,20]. In addition, some wanted to leave the profession or change their area of work to a department where there was less risk of experiencing traumatic events [7,10,21–23].

To enhance maternal and newborn quality care, it is therefore important to explore the experiences and perceptions of maternity care providers to better understand the traumatic

impact of work-related serious events on professional behaviour and their personal well-being. The aim of this international study was to examine the factors related to the impact of maternity care providers' involvement in traumatic events. The factors explored include defining the characteristics of the maternity care providers who participated in the study, to examine the type and frequency of severe traumatic events they were involved in/or witnessed, to identify the availability and use of formal and informal support, to identify the impact on their professional behaviour and work/career- related responses and report the impact on their professional Quality of Life (compassion satisfaction, burn-out) following traumatic events and examine if they experienced secondary traumatic stress.

## Methods

An international survey of maternity care professionals on MATernity serious EventS (MATES) was developed and hosted on Qualtrics[XM]. The MATES survey was developed following an extensive systematic literature review on validated instruments measuring serious events and outcomes of serious events among maternity healthcare providers. Following critical discussion, the research team decided on the range of validated instruments to be included, these were: Beck *et al.*, [15], Bride *et al.*, [24], Stamm, [25]; Toohill *et al.*, [26]. The research team (including maternity multidisciplinary providers and women representatives) then undertook to establish the content validity of the new instrument, and the questionnaire was piloted. Changes were not required following this process.

The finalised questionnaire included seven parts (P 1–7): P1—Demographic characteristics; P2—An 8 item, 4-point scale on the frequency and type of serious event experienced, and if the event had been subjected to an audit/report/legal case, yes/no; P3—A 4 item, 3-point scale of Formal and Informal Supports available and utilised; P4—A 4 item, 10-point visual analogue scale (VAS) on impact related to professional behaviour in terms of confidence and concerns; P5—Three questions on work impact, yes/no; P6—A 20 item, 5-point Likert scale relating to Professional Quality of Life with subscales compassion, satisfaction and burn out; P7—A 17 item, 5-point Likert scale related to personal well-being when recalling the serious event and present feelings.

From the outset the underpinning philosophy informing the research team was that it is the individual who experiences or witnesses the traumatic event(s) who decide if the event(s) was traumatic or not.

## Data Collection

The data were collected between 1st July and 31st December 2022 by disseminating the research recruitment flyer with the online anonymous Qualtrics[XM] survey link to 33 countries via the COST Action (CA18211) network and members social media accounts.

## Ethics

Ethical Approval was obtained from Zuyderland-Zuyd University, the Netherlands (7th December 2021; METCZ20210170). Potential respondents entered the survey through an information page explaining the study, where they gave their approval via ticking a check box relating to an online statement of consent, indicating that they understood the information and agreed to participate in the study. This action was a requirement (set within Qualtrics[XM]) to enabled respondents access to complete the survey, allowing respondents sufficient reflection time.

## Data analysis

In undertaking the analyses, response items were grouped to facilitate the large number of variables. The variable 'professional background' was grouped as 'midwives/nurses'

and 'physicians'. The variable 'type of serious event' was grouped as 'affecting infant' (items 1, 2, 3, 4), 'affecting mother' (items 5, 6) and 'aggression by woman or family' (item 7).

Univariate testing of potential explanatory variables associated with any of the four outcome scores (ProQOL-subscale Compassion Satisfaction, ProQOL- subscale Burnout, STS-scale at the time and STS-scale presently) was carried out using the unpaired t-test, one-way analysis of variance or Pearson correlation test, depending on the coding of the variable in question. This was followed by multivariate linear regression analyses, using a forward stepwise technique, to identify the main predictors, and adjust for any confounding effects. Data analyses were conducted using SPSS v.20. Two-sided significance tests were applied ($\alpha < .05$).

## Results

In total, 579 maternity care providers from 33 countries from Asia, Africia, Australia, Europe and North America participated in the study (See S1 File for list of countries). The questionnaire was completed fully by 399 respondents, while another 180 only answered the questions on being involved in serious events when caring for women and birthing people during their maternity care.

Respondents had various professional backgrounds including midwives (416), nurses (41), physicians (obstetricians (78), neonatologists (15), GPs (1)) plus other professionals (7) such as, mental health care providers (13) who cared for women during pregnancy, birth, and postnatal periods. Eight respondents did not mention their primary professional role. The age range of the respondents was from 20 to over 60 years, while their years of work experience ranged from less than five years to over 25 years. On average, the responders referred to a serious event that took place 5.33 years ago with a range from 10 days to 35 years. Demographic characteristics of respondents (including primary professional role) are presented in S1 File.

### Experiences with serious events

Respondents were invited to score the serious events with which they were directly involved in (Table 1). The events most often scored were '*Infant being born with a life-threatening condition*' and '*Severe or life-threatening maternal morbidity during pregnancy or birth*'. Nearly 65%

**Table 1. Types and frequencies of serious events (n = 579).**

| Type of serious event | never n (%) | once n (%) | twice n (%) | ≥ three n (%) |
|---|---|---|---|---|
| Infant death during birth | 295 (50.95) | 131 (22.63) | 56 (9.70) | 97 (16.75) |
| Neonatal death from birth-related causes | 316 (54.58) | 120 (20.73) | 43 (7.43) | 100 (17.27) |
| Infant born with life-threatening condition | 117 (20.21) | 101 (17.44) | 80 (13.82) | 281 (48.53) |
| Infant severely injured during birth | 259 (44.73) | 139 (24.01) | 70 (12.09) | 111 (19.17) |
| Maternal death during pregnancy or birth | 386 (66.67) | 122 (21.07) | 38 (6.56) | 33 (5.70) |
| Severe or life-threatening maternal morbidity during pregnancy or birth | 147 (25.39) | 114 (19.69) | 92 (15.89) | 226 (39.03) |
| Exposure to aggression by the woman or her family member | 205 (35.41) | 97 (16.75) | 91 (15.72) | 186 (32.12) |
| Exposure to any other severe event, such as … | 390 (67.36) | 62 (10.71) | 34 (5.87) | 93 (16.06) |

of respondents indicated that they had been exposed to violence or aggression by the woman or her family on one or more occasion.

To the open-ended question on other serious events, 128 responders mentioned a wide range of events, such as *Mother-related emergencies* (e.g., maternal suicide, post-partum haemorrhage (PPH)), *Fetus-related emergencies* (e.g., perinatal death), *Disrespect towards women* (e.g., obstetric violence, discrimination), *Women's complex social circumstances* (e.g., domestic violence, rape) and *Maternity care providers working experiences* (e.g., colleague's suicide, bullying) (S2 File).

## Availability and use of support by Maternity Care Provider

Respondents reported that they relied mainly on their partner, family, or friends for support after a serious event, which was available for around 85% of them. Institutional support programs were not available for almost 60% of the respondents and if available, were only used by 37% (Table 2). Almost 9% of the respondents (n = 40) had none of the mentioned support available.

## Professional and personal responses following serious events

About 20% (n = 92) of respondents reported that they had 'taken time off sick' or 'changed their professional allocation' (n = 81) after being involved in a serious event and 41.57% (n = 175) seriously considered leaving their position in maternity care (Table 3). In total, 51.78% (n = 218) of respondents reported at least one of these three responses.

We used the Professional Quality of Life Scale (ProQOL-Scale) to measure professional well-being of respondents recalling their feelings at the time of the event. Almost 65% of respondents derived limited pleasure from their work in the time after a serious event (with moderate to low scores on the ProQOL-subscale compassion satisfaction). Nearly 80% scored moderate to high on the ProQOL- subscale burnout, indicating they felt less effective in their work at the time (Table 4).

We used the Secondary Traumatic Stress Scale (STS-Scale) to measure the personal wellbeing of the respondents, recalling their feelings of wellbeing at the time of the event and their present feelings. The average score for the group was higher when recalling their feelings at the time compared to their present feelings (43.93 vs 33.66, respectively). In addition, over 21% had high to severe scores on the STS-Scale when currently thinking back on the serious event

**Table 2. Availability and use of support after a serious event (n = 469).**

| Support | Available and used n (%) | Available and not used n (%) | Not available n (%) |
|---|---|---|---|
| Institutional support programme | 71 (15.14) | 120 (25.59) | 278 (59.28) |
| Immediate supervisor | 198 (42.22) | 118 (25.16) | 153 (32.62) |
| Partner or family | 311 (66.31) | 82 (17.48) | 706 (16.20) |
| Friends | 283 (60.34) | 116 (24.73) | 70 (14.93) |

**Table 3. Work/career-related responses after a severe event (n = 421).**

| Response | Yes n (%) |
|---|---|
| Taken time off sick | 92 (21.85) |
| Change in professional allocation | 81 (19.24) |
| Seriously considered leaving the maternity care profession | 175 (41.57) |

(Table 5). In a sub-analysis we looked at the group for whom the event was less than a year ago. In this group (n = 51), 15 respondents scored high or severe. With 29%, this is slightly higher than the 21% in the total group.

## Factors associated with professionals and personal well-being

We conducted univariate analyses to explore associations between professional and personal wellbeing and various factors. These were: professional background (midwives/nurses or physicians); type of serious event (affecting infant or mother, aggression by woman or family); gender (male or female) and being subjected to investigation (via audit, reported to the regulatory authority, a complaint/involved in legal action). Other factors included: work/career related responses (sick, change allocation, consider leaving); time since event; age; years of experience, and confidence and concerns about advising and providing care (Table 6). There was no significant difference in mean scores on ProQOL-subscales and STS-scales between midwives/nurses and physicians, or between those who were and were not involved in serious events affecting infant or mother or time since the event or age.

A significant correlation was found with scores on both ProQOL-subscales and STS-scales for years of experience, confidence and concerns about advising and providing care, and work/career-related responses. More years of experience and higher confidence had positive correlations with Compassion Satisfaction and negative correlations with Burnout and Traumatic Stress, while higher concerns about advising and providing care had negative correlations with Compassion Satisfaction and positive correlations with Burnout and Traumatic Stress. Those taking time off sick, changing professional allocation or considering leaving

**Table 4.** Professional wellbeing after a serious event (n = 421).

| ProQOL-subscales* | mean | SD | Low n (%) | Moderate n (%) | High n (%) |
|---|---|---|---|---|---|
| Compassion Satisfaction | 37.95 | 7.30 | 13 (3.09) | 267 (63.42) | 141 (33.49) |
| Burnout | 27.28 | 6.14 | 89 (21.14) | 327 (77.67) | 5 (1.19) |
| **Confidence and concerns**** | mean | SD | | | |
| Confidence advising woman after event | 7.79 | 2.27 | | | |
| Confidence providing care after event | 7.57 | 2.48 | | | |
| Concerned advising woman after event | 4.39 | 3.14 | | | |
| Concerned providing care after event | 4.76 | 3.16 | | | |

*Possible range sum score per subscale: 10–50. Low: sum score 22 or less, Moderate: sum score between 23 and 41, High: sum score 42 or more for each subscale (Stamm, 2009).

**Continuous score from 1–10 with 1 = unconfident/not worried and 10 = confident/extremely worried

**Table 5.** Personal wellbeing after a serious event (n = 399).

| STS-Scale | mean | SD | Little or no* n (%) | Mild* n (%) | Moderate* n (%) | High* n (%) | Severe* n (%) |
|---|---|---|---|---|---|---|---|
| Total score for recalling feelings at the time of event | 43.93 | 14.66 | 56 (13.66) | 85 (20.73) | 65 (15.85) | 58 (14.15) | 135 (37.71) |
| Total score for present feelings | 33.67 | 12.87 | 140 (35.09) | 121 (30.33) | 55 (13.78) | 26 (6.52) | 57 (14.29) |

*Possible range total sum score 17–85. Little or no STS (sum score < 28), mild STS (sum score 28–37), moderate STS (sum score 38–43), high STS (sum score 44–48), and severe STS (sum score ≥49) (Bride, 2007).

**Table 6. Univariate exploration of factors associated with professional and personal wellbeing.**

| | ProQOL-subscale Compassion Satisfaction | | ProQOL-subscale Burnout | | STS-scale at the time | | STS-scale presently | |
|---|---|---|---|---|---|---|---|---|
| | mean | p-value | mean | p-value | mean | p-value | mean | p-value |
| **Professional background** | | | | | | | | |
| Midwives/nurses | 37.71 | 0.179 | 27.13 | 0.312 | 44.44 | 0.250 | 32.75 | 0.766 |
| Physicians | 38.99 | | 27.93 | | 42.19 | | 33.29 | |
| **Type of serious event** | | | | | | | | |
| Event effecting infant | | | | | | | | |
| no | 38.37 | 0.155 | 26.95 | 0.181 | 44.67 | 0.357 | 33.00 | 0.655 |
| yes | 37.29 | | 27.79 | | 43.29 | | 32.37 | |
| Event effecting mother | | | | | | | | |
| no | 38.19 | 0.592 | 27.11 | 0.650 | 43.80 | 0.703 | 32.19 | 0.506 |
| yes | 37.80 | | 27.39 | | 44.37 | | 33.12 | |
| Aggression by woman or family | | | | | | | | |
| no | 37.89 | 0.654 | 26.32 | 0.006 * | 43.30 | 0.065 | 31.14 | 0.106 |
| yes | 37.46 | | 28.59 | | 46.93 | | 34.24 | |
| **Gender** | | | | | | | | |
| Identified as male | 40.52 | 0.032 * | 26.59 | 0.482 | 39.79 | 0.058 | 33.03 | 0.767 |
| Identified as female | 37.84 | | 27.29 | | 44.38 | | 33.75 | |
| **Being subject to** | | | | | | | | |
| Audit yes | 37.34 | 0.046 * | 27.77 | 0.059 | 45.80 | 0.008 * | 33.85 | 0.066 |
| no | 38.73 | | 26.65 | | 41.99 | | 31.33 | |
| Report to regulatory authority yes | 37.10 | 0.081 | 28.39 | 0.007 * | 44.94 | 0.401 | 32.73 | 0.971 |
| no | 38.44 | | 26.64 | | 43.67 | | 32.78 | |
| Complaint/ legal action yes | 37.54 | 0.355 | 27.95 | 0.072 | 45.75 | 0.073 | 34.58 | 0.031 * |
| no | 38.21 | | 26.85 | | 43.10 | | 31.57 | |
| **Work/career-related responses** | | | | | | | | |
| Taken time of sick | | | | | | | | |
| yes | 33.97 | <0.001 * | 31.05 | <0.001 * | 53.85 | <0.001 * | 40.34 | <0.001 * |
| no | 39.06 | | 26.22 | | 41.38 | | 30.60 | |
| Change professional allocation yes | 33.21 | <0.001 * | 32.06 | <0.001 * | 54.18 | <0.001 * | 39.75 | <0.001 * |
| no | 39.08 | | 26.14 | | 41.75 | | 31.09 | |
| Consider leaving maternity care yes | 34.40 | <0.001 * | 30.65 | <0.001 * | 51.74 | <0.001 * | 37.62 | <0.001 * |
| no | 40.48 | | 24.89 | | 38.66 | | 29.25 | |
| **Support received from Institutional program** | | | | | | | | |
| Available and used | 38.97 | 0.201 | 27.12 | 0.797 | 45.18 | 0.797 | 33.10 | 0.554 |
| Available and not used | 37.02 | | 27.62 | | 43.67 | | 33.86 | |
| Not available | 38.09 | | 27.17 | | 44.07 | | 32.15 | |
| **Immediate supervisor** | | | | | | | | |
| Available and used | 38.61 | 0.244 | 26.80 | 0.380 | 44.11 | 0.796 | 32.16 | 0.293 |
| Available and not used | 37.71 | | 27.60 | | 43.46 | | 31.84 | |
| Not available | 37.24 | | 27.67 | | 44.76 | | 34.33 | |
| **Partner or family** | | | | | | | | |
| Available and used | 37.61 | 0.380 | 27.76 | 0.059 | 45.41 | 0.043 * | 33.79 | 0.036 * |
| Available and not used | 38.77 | | 25.96 | | 41.40 | | 29.04 | |
| Not available | 38.52 | | 26.67 | | 41.70 | | 32.39 | |

*(Continued)*

**Table 6.** (Continued)

| | ProQOL-subscale Compassion Satisfaction | | ProQOL-subscale Burnout | | STS-scale at the time | | STS-scale presently | |
|---|---|---|---|---|---|---|---|---|
| **Friends** | | | | | | | | |
| Available and used | 37.23 | 0.011 * | 27.80 | 0.052 * | 46.38 | 0.001 * | 33.87 | 0.121 |
| Available and not used | 38.39 | | 26.92 | | 41.18 | | 31.35 | |
| Not available | 40.26 | | 25.75 | | 40.02 | | 30.50 | |
| | correlation | p-value | correlation | p-value | correlation | p-value | correlation | p-value |
| **Age** | 0.09 | 0.074 | −0.10 | 0.059 | −0.04 | 0.470 | −0.08 | 0.133 |
| **Years of work experience** | **0.17** | **0.001 *** | **−0.16** | **0.001 *** | **−0.10** | **0.047 *** | **−0.11** | **0.023 *** |
| **Time since event** | −0.05 | 0.531 | −0.03 | 0.717 | 0.05 | 0.460 | −0.10 | 0.149 |
| **Confidence advising woman after event** | **0.34** | **<0.001 *** | **−0.25** | **<0.001 *** | **−0.28** | **<0.001 *** | **−0.17** | **<0.001 *** |
| **Confidence providing care after event** | **0.49** | **<0.001 *** | **−0.38** | **<0.001 *** | **−0.39** | **<0.001 *** | **−0.18** | **<0.001 *** |
| **Concerned advising woman after event** | **−0.17** | **0.001 *** | **0.24** | **<0.001 *** | **0.23** | **<0.001 *** | **0.13** | **0.008 *** |
| **Concerned providing care after event** | **−0.20** | **<0.001 *** | **0.28** | **<0.001 *** | **0.29** | **<0.001 *** | **0.19** | **<0.001 *** |

*significant due to p <= 0.05.

maternity care had lower mean scores on Compassion Satisfaction and higher scores on Burnout and Traumatic Stress.

Significant differences in mean scores on ProQOL-subscale Compassion Satisfaction were found for gender, being subjected to investigation and support received from friends, with female respondents having a lower mean score (37.84 versus 40.52) similar to those subject to an audit (37.34 versus 38.73) and those using support from friends.

On the ProQOL-subscale Burnout, those experiencing aggression had significantly higher mean score (28.59 versus 26.32), similar to those who were subject to report to regulatory authorities (28.39 versus 26.64) and those using support from friends. STS scores at the time of the event were higher for those being subject to an audit after a serious event (45.80 versus 41.99), and those using support from partner or family and friends. There was a trend with higher scores for those who identified as female (44.38 versus 39.79; p-value = 0.058). STS scores presently were higher for those being subject to complaint/legal action after a serious event (34.85 versus 31.57) and those using support from partner or family.

In Table 7, multivariate linear regression analyses showed that higher scores on the ProQOL-subscale Compassion Satisfaction are significantly correlated with more years of work experience and more confidence in providing care after the event. Those experiencing aggression by the woman or her family and being more concerned about providing care after the event are significantly associated with an increase in the ProQOL-subscale Burnout. Therefore, experiencing aggression by the woman or her family increases the level of MCP burnout. More years of work experience and more confidence in providing care after the event are significantly associated with a decrease on the same ProQOL-subscale. Thereby the results suggests that more years of experience decreases the MCP's level of burnout and their level of Secondary Traumatic Stress.

Support from friends is significantly associated with an increase in the score on the STS-scale at the time of the event compared to when support is not available. However, when support is available and not utilised, a lower score on the STS-scale is found compared to when support is just not available. More confidence in providing care after the event is significantly associated with a lower STS score at the time. Conversely being more concerned about providing care after the event is significantly associated with an increased STS score. In terms of the

**Table 7. Multivariate linear regression of factors associated with professional and personal wellbeing.**

| | ProQOL-subscale Compassion Satisfaction | | ProQOL-subscale Burnout | | STS-scale at the time | | STS-scale presently | |
|---|---|---|---|---|---|---|---|---|
| | Reg Coeff | p-value | Reg Coeff | p-value | Reg Coeff | p-value | Reg Coeff | p-value |
| **Type of serious event** | | | | | | | | |
| Aggression by woman or family no | | | 0 | <0.001 * | 0 | <0.001 * | | |
| yes | | | 3.26 | | 5.26 | | | |
| **Support received from Partner or family** | | | | | | | | |
| Available and used | | | | | | | 0.54 | 0.029 * |
| Available and not used | | | | | | | −4.15 | |
| Not available | | | | | | | 0 | |
| **Friends** | | | | | | | | |
| Available and used | | | | | 2.68 | 0.010 * | | |
| Available and not used | | | | | −1.70 | | | |
| Not available | | | | | 0 | | | |
| **Years of work experience** | 0.58 | 0.001 * | −0.65 | 0.001 * | | | −0.87 | 0.024 * |
| **Confidence providing care after event** | 1.40 | <0.001 * | −0.80 | <0.001 * | −2.07 | <0.001 * | −0.69 | 0.013 * |
| **Concerned providing care after event** | | | 0.36 | <0.001 * | 0.73 | <0.001 * | 0.69 | 0.001 * |

* significant due to p <= 0.05.

present STS-scale, both more years of work experience and more confidence about providing care after the event are significantly associated with a drop in the STS scale, whereas being more concerned about providing care after the event is significantly associated with a rise. Utilising support from one's partner or family is significantly associated with a higher score on the STS-scale, whereas having help available but choosing not to use it, is associated with a significant lower score on STS-scale at present, when compared to not having any help available from one's partner or family.

## Discussion

In this international study, we explored the different factors associated with maternity care providers' experiences of work-related serious events and the subsequent impact on their practice and personal well-being. The respondents included: midwives, nurses, physicians, obstetricians, neonatologists, GPs, and mental health care providers. In total, there was 579 respondents, representing 33 different countries. The response rate is similar to other studies [7,27], which examined exposure of severe events among individual MCPs such as midwives and obstetricians however our study gained experiences from a wider range of MCPs. On average maternity care providers reflected on serious events that occurred five years previously, while some recalled events up to 35 years and as little as 10 days prior to completing the questionnaire.

Our findings demonstrate that many MCPs are exposed to serious events during their work, with the type of serious event(s) varying as illustrated in Table 1 and S2 File. More than 75% of MCPs stated that the serious event/s to which they had been exposed to, posed a risk to the mother's or infant's lives. This compares with Wahlberg *et al.*, [6] findings, who also highlighted that 71% midwives and 84% obstetrician in Sweden, experienced severe obstetric event(s) which had the potential to have life-threatening consequences for the woman or the

new-born. Therefore, this phenomenon is not unique to anyone jurisdiction given that this study had an international sample.

Our study found that notwithstanding respondent's professional discipline background, being exposed to serious events had adverse effects equally on both midwives and physicians. The MCP respondents were negatively affected and experienced secondary traumatic stress and burnout. Systematic reviews by Bingham *et al.,* [28] and Aydin and Aktaş [23] on the impact of midwives exposed to traumatic birth(s) concur, highlighting how they struggled personally and professionally, often feeling as the '*Forgotten victims'* [28]. This study thereby, further demonstrates the devastating effect on MCPs, as approximately a fifth of respondents had to either 'taken time off sick' or 'changed their professional allocation', with over 40% of MCPs seriously considered leaving their profession. In a time of international reduced maternity care staffing, the effect of serious events on MCPs is an increase in the attrition rate and a cause of professional shortages [29,30].

Other consequences of traumatic birth on MCPs responders are the effect on their future practice following a serious event(s). Our study highlighted that over 65% (n = 280) of MCPs after a serious event experienced limited pleasure from their work and recorded moderate to low compassion scores on the ProQOL-subscale, with 80% scoring moderate to high in relation to burnout. Moderate to low compassion scores held by a MCP will ultimately have an effect on maternity care users and their future practice. MCPs may enhanced their practice positively through reflection of the serious event and subsequently experience personal and professional development [26,28,31]. Other MCPs tend to practice defensively, losing their trust in the physiological birth process [14] and intervene (for example: amniotomy, induction of labour or caesarean section) which may not benefit the infant or woman [32,33], and even do harm in the short and long-term [34,35]. Smith [36] agrees that fear can become prevalent in MCPs following serious events and reflected in an increasing culture of risk adverse practice.

As highlighted the work-related serious events were not just medical related issues happening to the mother and/or infant, two thirds of MCP were exposed, on one or more occasion to violence or aggression by the woman or her family. Outbursts of aggression from the woman towards MCPs are perhaps more understanding, when the woman is experiencing strong uterine contractions in labour, nevertheless violence and aggression towards MCPs is disrespectful and not acceptable with Zero Tolerance policies [37]. These findings demonstrate the serious impact on MCPs from being frequently subjected to unacceptable behaviour as part of their work. Employing de-escalation strategies like those highlighted by NICE [38] are essential in this situation. It is important to note that MCPs in this study who had increased years of work experience and confidence in providing care after the serious event was protective from them experiencing burnout and secondary traumatic stress in the long term.

Having experienced these work-related serious events, a high proportion of respondents indicated that they relied mainly on their partner, family, or friends for support. Crucially, the respondents who sought this support also had a higher score on burnout, suggesting that these MCPs were impacted enough by the serious events to open and discuss their experiences with their family and friends. Providing this support must also be challenging for family and friends, as it may leave them worried or anxious for their loved ones. Disappointingly, over two thirds of respondents indicated institutional support programs were not available and if they were available, were only used by one third, with a small minority reporting that no support was available. These findings are like previous studies synthesised by Nieuwenhuijze *et al.* and suggests that despite growing evidence of the need for support, health service providers are either not providing support desired or support that appropriately meets MCPs needs [39].

The lack of formal support and engagement from health service providers with MCPs experiencing such events ultimately will lead to loss of revenue for maternity care organisations. The impact our international research highlights that if we don't care and provide effective support for MCP they will continue to leave. Nieuwenhuijze *et al.* calls for serious consideration of how maternity care organisations can maintain MCPs wellbeing through positive engagement while they undertake their MCP role [39].

## Conclusion

The findings from this international survey demonstrates the impact of being involved in maternity serious events on MCPs individual personal and professional lives. Many MCPs experienced aggression, burnout, secondary traumatic stress, lacked compassion, had to take sick leave, moved their area of work and/or ultimately considered leaving their profession. With international shortages of MCPs the impact of these events is contributing to this shortage.

Women and their families will also have experienced the impact of MCPs being involved in maternity serious events with a demonstration of a lack of compassion and satisfaction within their role. Women and babies may also be exposed to unnecessary maternity care interventions as MCPs potentially undertake risk adverse practice because of work-related fears.

In the absence of institutional support, many MCPs had to rely on informal support from partner, family and friends. Further research is required to understand why informal support is most often used. Management within maternity care health systems need to identify and implement effective support methods for MCPs to address the impact of exposure to serious events and protect their wellbeing. Through effective leadership, psychological safety can be created, learning undertaken and the prevention of avoidable errors [36]. Dignity and respect are important in the care of women, but it is also essential for MCPs who are often exposed to serious events within their work, leaving them affected and distressed. Part of sustainable maternity care provision is when health systems value and protect their workforce [32]

The strength of this research is that MCPs from a large number of countries participated (n = 33) in the study. In addition, validated instruments were incorporated within the questionnaire design [15,24–26], content validity was undertaken, and the questionnaire was piloted. There was however a large variation in the number of people who participated from different countries. Therefore, we could not explore the influence of culture and organisation of maternity care as a contributing factor. We also did not ask about support received from colleagues (we simply forgot to put this in). Colleagues might be the right mixture of formal and informal support that provides an answer to dealing with traumatic events, as long as it is non-judgemental and provided within a safe environment [40].

## Recommendations/Implications for practice/ Future Research/ Limitations

There is an urgent need to provide institutional evidenced based formal supports for all MCPs. Programmes are being piloted [41] and following detailed evaluation of the impact, consideration regarding implementation is necessary to provide an effective peer support system for MCPs. Where formal support exists, research is required to explore why it is not being used and why informal support has little impact. It is also important to explore de-escalation technique training and co-produce policies with MCPs for de-escalating aggressive behaviour of adults. In addition, we recommend regular interprofessional education (IPE) and multi-disciplinary meetings between MCPs utilising simulated learning to support evidence-based practice during serious events.

Furthermore, research is necessary to explore why MCPs experience aggression from partners and what the mechanism is to prevent and address this unacceptable behaviour. Research is also required to investigate the impact that work related serious events has on MCPs professional behaviour, including sick leave and/or leaving their profession. It is important to examine what respectful care to women and maternity care providers can achieve. Furthermore, research into why MCPs tend to seek support from family and friends is necessary, and the subsequent impact on family and friends on providing this informal support.

A limitation of the study is that the survey was disseminated in English language to 33 countries via COST Action members and the results are also published in English.

## Supporting information

**S1 File. Demographic characteristics of respondents.**
(XLSX)

**S2.File. Serious events.**
(DOCX)

## Acknowledgements

This paper is based upon work from COST Action CA18211 Perinatal Mental Health; Birth Trauma: Maximising Best Practice; Optimal Outcomes https://www.ca18211.eu/. We also acknowledge all the members of WG2 of this COST Action for their work on data collection.

## Author contributions

**Conceptualization:** Maria Healy, Patricia Leahy-Warren, Joan Lalor, Eleni Hadjigeorgiou, Marianne Nieuwenhuijze.

**Data curation:** Maria Healy, Jean Calleja-Agius, Neville Calleja, Marianne Nieuwenhuijze.

**Formal analysis:** Maria Healy, Jean Calleja-Agius, Neville Calleja, Marianne Nieuwenhuijze.

**Investigation:** Maria Healy, Patricia Leahy-Warren, Eleni Hadjigeorgiou.

**Methodology:** Maria Healy, Patricia Leahy-Warren, Jean Calleja-Agius, Marianne Nieuwenhuijze.

**Project administration:** Maria Healy, Marianne Nieuwenhuijze.

**Software:** Maria Healy, Jean Calleja-Agius, Joan Lalor.

**Writing – original draft:** Maria Healy, Patricia Leahy-Warren, Jean Calleja-Agius, Neville Calleja, Joan Lalor, Eleni Hadjigeorgiou, Marianne Nieuwenhuijze.

**Writing – review & editing:** Maria Healy, Patricia Leahy-Warren, Jean Calleja-Agius, Neville Calleja, Joan Lalor, Eleni Hadjigeorgiou, Marianne Nieuwenhuijze.

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
