## [Decision Letter · Decision Letter 0]

9 Sep 2024

PONE-D-24-17808MATernity care providers’ experiences of work-related serious EventS (MATES): An International surveyPLOS ONE

Dear Dr. Healy,

Thank you for submitting your manuscript to PLOS ONE. After careful consideration, we feel that it has merit but does not fully meet PLOS ONE’s publication criteria as it currently stands. Therefore, we invite you to submit a revised version of the manuscript that addresses the points raised during the review process.

Dear Dr Healy,

Your submission requires minor revisions. Both reviewers have recommended minor revisions. Therefore, I invite you to respond to the reviewers' comments and submit your revised your manuscript.

We look forward to receiving your revised manuscript.

Kind regards,

Sunita Panda, PhD

Academic Editor

PLOS ONE

Reviewers' comments:

Reviewer's Responses to Questions

**Comments to the Author**

1. Is the manuscript technically sound, and do the data support the conclusions?

Reviewer #1: Yes

Reviewer #2: Yes

2. Has the statistical analysis been performed appropriately and rigorously? 

Reviewer #1: I Don't Know

Reviewer #2: Yes

3. Have the authors made all data underlying the findings in their manuscript fully available?

Reviewer #1: Yes

Reviewer #2: Yes

4. Is the manuscript presented in an intelligible fashion and written in standard English?

Reviewer #1: Yes

Reviewer #2: Yes

5. Review Comments to the Author

Reviewer #1: This is a useful survey and provides important information on the experiences of maternity care providers. Just some issues to address:

L 98 -100 The term 'positive experience' comes from the WHO (2018) which states that childbirth should be a positive life event to optimize the experience of childbirth for women and their babies while recognising that adverse events can and do occur. The context of this statement should be clarified which could be followed by the Beck reference.

If this is an international study, perhaps the authors could present international data on adverse events (morbidity/mortality) – these are expectations for MCPs globally.

L178. I presume that informed consent was by means of a check box to proceed with the online study, did the participants also submit a 'written consent'? Also, explain what is meant by the term 'sufficient reflective time' for an online survey.

L 201. Were the 180 incomplete surveys included in the analysis?

Provide more information on the 33 countries? Europe, Asia, Africa, Americas?

Also a breakdown of the 353 participants would be helpful, numbers of midwives, nurses, physicians, obstetricians, neonatologists, GPs, and mental health care providers. It seems most respondents nurse/midwives or physicians.

Consider the limitations - survey in English excluded potential participants, also bias in relation to publications in English (US, Australia, UK, Ireland, Nordic countries).

Reviewer #2: The aim of this international study was to examine the factors related to the impact of

maternity care providers’ involvement in traumatic events. An anonymised online survey relating to MATernity serious EventS (MATES) was developed utilising validated instruments and hosted on QualtricsXM across 33 countries between 1st July and 31st December 2022 via social

media and the COST Action network. In total, 579 MCPs responded with a wide age range and years of experience.

Findings suggest that MCPs are exposed to a large variation of serious events and

continue to be intensely affected, up to the present day. Events ranged from stillbirth,

neonatal death, maternal death, severe or life-threatening maternal or infant incidents and

violence and aggression from women or family member. Institutional support for staff is

limited, and when available, seldom used. Family and friends were relied on for support,

but this does not appear to be associated with MCPs experiencing less secondary traumatic stress. Subsequently MCPs were absence from work through sickness (22%),

changed their professional allocation (19%) and seriously considered leaving (42%).

Moreover, many reported low to moderate compassion satisfaction and burnout (65-

80%). With international scarcity of MCPs, the impact of these events seem to contribute to this

shortage. Effective support for MCPs is required if staff are to be recruited and retained.

This is an important study and the strengths and limitations of the method have been described.

It is an important finding that nearly 65% of respondents indicated that they had been exposed to violence or aggression by the woman or her family on one or more occasion.

It is also important that respondents reported that they relied mainly on their partner, family, or friends for support after a serious event, and that Institutional support programs were not available for almost 60% of the respondents and if available, were only used by 37%. The reasons for lack of access needs to be explored in the future.

Table 7 is complex and worth explaining in words the key relationships and implications from the table. In terms of future actions the authors may wish to inform themselves of the Trauma Risk Management (TRiM) TRIM programme which is being piloted in healthcare as a potential action to provide early support for colleagues.

D. Whybrow, N. Jones, N. Greenberg, Promoting organizational well-being: a comprehensive review of Trauma Risk Management, Occupational Medicine, Volume 65, Issue 4, June 2015, Pages 331–336, https://doi.org/10.1093/occmed/kqv024

6. PLOS authors have the option to publish the peer review history of their article (what does this mean? ). If published, this will include your full peer review and any attached files.

**Do you want your identity to be public for this peer review?** For information about this choice, including consent withdrawal, please see our Privacy Policy .

Reviewer #1: No

Reviewer #2: **Yes: ** Jane Sandall

---

## [Author Response · Author response to Decision Letter 0]

18 Oct 2024

Response to Reviewers

Reviewer #1:

L 98 -100 The term 'positive experience' comes from the WHO (2018) which states that childbirth should be a positive life event to optimize the experience of childbirth for women and their babies while recognising that adverse events can and do occur. The context of this statement should be clarified which could be followed by the Beck reference.

Thank you for this comment, see amended sentences:

The World Health Organization (WHO) stresses that a positive childbirth experience should be the ultimate goal of maternity care for all women, even when medical interventions are necessary [1]. However, serious complications can occur throughout the perinatal period, with most happening during labour and birth [2].

Reviewer 1 Comment

If this is an international study, perhaps the authors could present international data on adverse events (morbidity/mortality) – these are expectations for MCPs globally.

Authors comment:

As this study focuses on maternity care providers’ experiences of work-related serious events and does highlight their risk of exposure (67%-90%) to a traumatic birth event. The authors suggest that relevant data is presented from the providers’ perspective.

Reviewer 1 Comment

L178. I presume that informed consent was by means of a check box to proceed with the online study, did the participants also submit a 'written consent'?

Also, explain what is meant by the term 'sufficient reflective time' for an online survey.

Authors comment - Thank you, we understand how these sentences require clarity.

No participants did not submit written consent, a check box was ticked to proceed with the online study. Participants could also return via the link to the online survey - See amended sentences in manuscript:

Potential participants entered the survey through an information page explaining the study, where they gave their approval via ticking a check box relating to an online statement of consent, indicating that they understood the information and agreed to participate in the study. This action was a requirement (set within QualtricsXM) to enabled participants access to complete the survey, allowing participants sufficient reflection time.

Reviewer 1 Comment

L 201. Were the 180 incomplete surveys included in the analysis?

Authors response: Thank you for seeking clarification. Yes, the 180 incomplete surveys were included, as despite these participants not completing the whole survey they did answer key questions on being involved in serious events during maternity care. In addition, all findings presented within the tables are reported with the n= value clearly highlighted.

Reviewer 1 Comment

Provide more information on the 33 countries? Europe, Asia, Africa, Americas?

Also a breakdown of the participants would be helpful, numbers of midwives, nurses, physicians, obstetricians, neonatologists, GPs, and mental health care providers. It seems most respondents nurse/midwives or physicians.

Authors response: Thank you for this comment – further detail is provided in the manuscript.

Manuscript now reads: In total, 579 maternity care providers from 33 countries from Asia, Africia, Australia, Europe and North America participated in the study (See S1 for list of countries).

Authors response - Manuscript now reads:

Respondents had various professional backgrounds including midwives (416), nurses (41), physicians (obstetricians (78), neonatologists (15), GPs (1)) plus other professionals (7) such as, mental health care providers (13) who cared for women during pregnancy, birth, and postnatal periods. Eight respondents did not mention their primary professional role.

Demographic characteristics of respondents (including primary professional role) are presented in S1.

Reviewer 1 Comment:

Consider the limitations - survey in English excluded potential participants, also bias in relation to publications in English (US, Australia, UK, Ireland, Nordic countries).

Authors response:

We agree with this comment – see additional insert on heading and sentence added.

Manuscript now reads: Recommendations/Implications for practice/ Future Research/Limitations

A limitation of the study is that the survey was disseminated in English language to 33 countries via COST Action members and the results are also published in English.

Reviewer #2: Comments

Table 7 is complex and worth explaining in words the key relationships and implications from the table.

Authors response: Thank you, see additional explanation provided

Manuscript now reads: Therefore, experiencing aggression by the woman or her family increases the level of MCP burnout. More years of work experience and more confidence in providing care after the event are significantly associated with a decrease on the same ProQOL-subscale. Thereby the results suggests that more years of experience decreases the MCP’s level of burnout and their level of Secondary Traumatic Stress.

Review 2 Comments:

In terms of future actions the authors may wish to inform themselves of the Trauma Risk Management (TRiM) TRIM programme which is being piloted in healthcare as a potential action to provide early support for colleagues. D. Whybrow, N. Jones, N. Greenberg, Promoting organizational well-being: a comprehensive review of Trauma Risk Management, Occupational Medicine, Volume 65, Issue 4, June 2015, Pages 331–336, https://doi.org/10.1093/occmed/kqv024

Authors Response:

Thank you for your suggestion – see amended sentence

Manuscript now reads: Programmes are being piloted [42] and following detailed evaluation of the impact, consideration regarding implementation is necessary to provide an effective peer support system for MCPs.

---

## [Decision Letter · Decision Letter 1]

3 Jan 2025

MATernity care providers’ experiences of work-related serious EventS (MATES): An International survey

PONE-D-24-17808R1

Dear Dr. Healy,

We’re pleased to inform you that your manuscript has been judged scientifically suitable for publication and will be formally accepted for publication once it meets all outstanding technical requirements.

Kind regards,

Sunita Panda, PhD

Academic Editor

PLOS ONE

Additional Editor Comments (optional):

Reviewers' comments:

Reviewer's Responses to Questions

**Comments to the Author**

1. If the authors have adequately addressed your comments raised in a previous round of review and you feel that this manuscript is now acceptable for publication, you may indicate that here to bypass the “Comments to the Author” section, enter your conflict of interest statement in the “Confidential to Editor” section, and submit your "Accept" recommendation.

Reviewer #1: All comments have been addressed

2. Is the manuscript technically sound, and do the data support the conclusions?

Reviewer #1: Yes

3. Has the statistical analysis been performed appropriately and rigorously? 

Reviewer #1: I Don't Know

4. Have the authors made all data underlying the findings in their manuscript fully available?

Reviewer #1: Yes

5. Is the manuscript presented in an intelligible fashion and written in standard English?

Reviewer #1: Yes

6. Review Comments to the Author

Reviewer #1: All recommendations have been adequately addressed by the authors, it is important that this paper is published.

7. PLOS authors have the option to publish the peer review history of their article (what does this mean? ). If published, this will include your full peer review and any attached files.

**Do you want your identity to be public for this peer review?** For information about this choice, including consent withdrawal, please see our Privacy Policy .

Reviewer #1: No

---

## [Editor Report · Acceptance letter]

PONE-D-24-17808R1

PLOS ONE

Dear Dr. Healy,

I'm pleased to inform you that your manuscript has been deemed suitable for publication in PLOS ONE. Congratulations! Your manuscript is now being handed over to our production team.

Kind regards,

on behalf of

Dr Sunita Panda

Academic Editor

PLOS ONE